# The Need for Breathing Training Techniques: The Elephant in the Heart Failure Cardiac Rehabilitation Room: A Randomized Controlled Trial

**DOI:** 10.3390/ijerph192214694

**Published:** 2022-11-09

**Authors:** Abeer Farghaly, Donna Fitzsimons, Judy Bradley, Magda Sedhom, Hady Atef

**Affiliations:** 1Department of Physical Therapy for Cardiovascular/Respiratory Disorder & Geriatrics, Faculty of Physical Therapy, Cairo University, Giza, Egypt; 2School of Nursing and Midwifery, Queen’s University of Belfast, Belfast, UK; 3Wellcome Trust-Wolfson NI Clinical Research Facility, Queen’s University Belfast, Belfast, UK; 4Basic Science Department, Faculty of Physical Therapy, Cairo University, Giza, Egypt

**Keywords:** breathing calisthenics, breathing training, cardiac rehabilitation, chronic heart failure, inspiratory muscle training

## Abstract

**Highlights:**

**What is missed?**
Standard cardiac rehabilitation (CR) programs do not typically consider respiratory symptoms.

**What are the main findings?**
Breathing exercises (BE) have a positive physiological effect on chronic heart failure (CHF).

**What are the implications of the main findings?**
Future CR programs for CHF have to manage respiratory symptoms.CR programs for CHF have to address patient-centered outcomes.

**Abstract:**

Background: Although solid evidence has indicated that respiratory symptoms are common amongst patients with chronic heart failure (CHF), state-of-the-art cardiac rehabilitation (CR) programs do not typically include management strategies to address respiratory symptoms. This study investigated the effect of the addition of breathing exercises (BE) to the CR programs in CHF. Methods: In a two parallel-arm randomized controlled study (RCT), 40 middle-aged patients with CHF and respiratory symptoms were recruited and randomized into two equal groups (*n* = 20); group (A): standard CR with BE and group (B): standard CR alone. Primary outcomes were respiratory parameters and secondary outcomes included cardiovascular and cardiopulmonary outcomes. All the participants attended a program of aerobic exercise (three sessions/week, 60–75% MHR, 45–55 min) for 12 weeks, plus educational, nutritional, and psychological counseling. Group (A) patients attended the same program together with BE using inspiratory muscle training (IMT) and breathing calisthenics (BC) (six sessions/week, 15–25 min) for the same duration. Results: There was a significant improvement in the respiratory outcomes, and most of the cardiovascular and cardiopulmonary outcomes in both groups with a greater change percentage in group A (*p* < 0.05). Conclusions: These results indicate that the addition of BE to the CR programs in CHF is effective and is a “patient-centered” approach.

## 1. Introduction

Chronic heart failure (CHF) is a common and debilitating condition that has cardiovascular (such as arrhythmias, elevated heart rate, and blood pressure [1]), cardiopulmonary (such as fatigue, declined exercise tolerance, and increased ventilatory demand during exercise [2]), and respiratory symptoms. The respiratory symptoms include dyspnea, chest pain, fatigue, shortness of breath, sleep apnea, and bronchitis-like symptoms. These symptoms add to the risk profile and detrimentally affect the quality of life [3].

In clinical practice, cardiac rehabilitation (CR) is considered standard care for CHF patients to mainly improve cardiovascular and/or cardiopulmonary manifestations [4]. Skeletal muscle abnormalities are highly prevalent in CHF and are associated with an increase in the ergoreflex, a muscle reflex stimulated by work performed [5]. Cardiac rehabilitation was proven to be effective in this context (reversal of the ergoreflex-induced muscle weakness) [6]. 

Systematic reviews have confirmed that addressing respiratory symptoms in CHF is crucial and affects the outcome of the treatment, and they recommended the blending of exercise training with BE to optimize outcomes. Typically standard CR programs do not include strategies to control respiratory symptoms [7,8].

The most successful state-of-the-art CR programs globally, such as the EuroAction [9], Our hearts Our minds [10,11], Coroprevention [12], and the Million hearts [13], include CHF among their target population. However, none of them consider BE as a method of controlling the well-known respiratory symptoms in CHF patients. 

Addressing these symptoms is important to CHF patients as it improves dyspnea. Improving ventilation in CHF would reduce ventricular preload and afterload, decrease extra-vascular lung water, and the work of breathing in CHF, improving many of the pathophysiologic mechanisms of the CHF [14,15]. Hence, an intervention that improves ventilation is needed, such as breathing training. 

Breathing training (BE) techniques that aim to improve the function of the respiratory muscles through specific exercises incorporating different methods like inspiratory Muscle Training (IMT) and breathing calisthenics (BC) [16]. IMT has been shown to improve respiratory muscle function and might help to reduce dyspnea on exertion [17]. It also improves heart rate variability, cardiac sympathovagal balance, and functional capacity [18]. BC is a series of breathing exercises performed to strengthen the diaphragmatic and abdominal muscles, improving respiration [19]. 

The objective of this study was to assess the effect of the addition of the BE techniques to the standard CR on the respiratory, cardiovascular, and cardiopulmonary symptoms in comparison with the standard CR alone, which is usually prescribed for CHF patients. 

## 2. Methods

### 2.1. Study Design and Setting

This study was a two-parallel arm RCT study. The recruitment period was 9 months, from June 2019 to March 2020. All patients signed a written consent before enrollment in the study. This study conforms to the principles of the Declaration of Helsinki and relevant ethical guidelines [20]. Ethical approval has been obtained from the Ethical committee of the faculty of Physical Therapy, Cairo University (approval number: P.T.REC/012/001906). This trial was registered on ClinicalTrials.gov, with the registration number NCT04905433, registered 27 May 2021.

### 2.2. Study Population

Forty patients (20 in each group) with CHF and respiratory symptoms were included in the study. The patients were in the age range between 45 and 65 years. The following were the eligibility criteria for the patients:

#### 2.2.1. Inclusion Criteria

Medically stable patients with CHF (≥1 year) and respiratory muscle weakness ≤ 70% of their predicted maximal MIP who were under optimized medical therapy for at least 3 months before entering the study. Patients with an ejection fraction ≤ 40% in NYHA class II and III and a stable condition (No rales on auscultation or tibial edema and with sinus rhythm) [21].

#### 2.2.2. Exclusion Criteria

Patients with chronic lung disorders, marked decline FVC, FEV1, FEV1/FVC < 70%, anemia, or severe hypoxia, resting dyspnea, uncontrolled arrythmias, or a history of myocardial infarction or pulmonary edema six months before the study. Additionally, patients with severe uncontrolled hypertension, uncontrolled diabetes mellitus, autonomic disorders, engagement in any regular physical training program for at least one month before the start of the study, or any neuromuscular disorder that may have hindered normal engagement in physical activity were excluded from the study.

### 2.3. Randomization and Grouping

The sealed envelope approach was used for randomization with an allocation ratio of 1:1. The two groups were the standard CR integration with breathing training group (group A) and the standard cardiac rehabilitation group (group B), with 20 participants in each group. The participants were not blinded to treatment allocation. 

### 2.4. Assessment

Before the initial assessment, all patients attended a comprehensive consultation with the multidisciplinary team (MDT) to be asked about their risk factors and physical activity level through the last month, and to address their chief complaints and motivations for the program. The assessments were conducted before the study and after 12 weeks (the study period). Figure 1 shows the design of the study.

#### 2.4.1. Pulmonary Function Testing (PFT): Spirometry System with Shutter (Zan Type 100-Germany)

A pulmonary function test was performed before and after the interventions. It was performed before the cardiopulmonary exercise testing (CPET) to measure forced vital capacity (FVC), forced expiratory volume in the first second (FEV1), and the ratio between them (FEV1/FVC), and MIP [22].

MIP was performed to measure the inspiratory strength before and after the study for both groups and to determine the starting load for the inspiratory muscle training group. Before the measurement, all the procedures were explained in detail to the patient. Therefore, the participant’s respiration had to be corrected verbally by giving a respiration rhythm. Following that, the patient sat in front of the spirometry and was instructed to first breathe normally, then exhale as slowly as possible almost to the residual volume of the lung through a clean sterilized mouthpiece connected to a shutter system. By pushing on the button of MIP (maximum inspiratory pressure), the maximum measurement was initiated, and by the next expiration, the flow was broken for 1 or 2 s. The patient was then instructed to inhale as deeply as possible; the highest pressure attainable was recorded as maximum inspiratory pressure (MIP) [23].

#### 2.4.2. Modified Borg Scale 

Dyspnea was evaluated during CPET using a modified Borg scale of 0 to 10; the patient documented the degree of dyspnea at rest and during the self-care activity of daily living [24]. 

#### 2.4.3. Twenty-Four Hour Ambulatory Electrocardiography (ECG) Recording (Holter)

We used three channels of ECG data that were put for the patient for 24 h for a full-day monitoring. The patients were asked to perform their activities of daily living and then sleep with the ECG channels kept in place. The resulting records were then analyzed by a Vision Premier Altair PC Halter system. The following parameters were calculated: Frequency domain parameters: LF (Hz): LF power in normalized units LF/(total power-VLF) × 100, HF-frequency (Hz), and LF/HF ratio [25]. HF peak is predominately related to the respiratory modulation of the cardiac vagal input. On the other hand, LF appears to represent a complex mixture of sympathetic and parasympathetic modulation of the heart rate, whereas the LF/HF is a reasonable measure of the relative effect of the sympathetic and parasympathetic outflows to the sinoatrial (SA) node [26]. 

Any recording of significant arrythmias and/or motion noise were excluded. 

#### 2.4.4. Cardiopulmonary Exercise Test (CPET)

Before conducting this symptoms-limited exercise test, the testing protocol was explained in detail to every patient before the initial assessment. Resting measurements (resting ECG by placing six chest electrodes and four limb electrodes on the patient, HR recorded by the ECG, SBP, and DBP recorded by sphygmomanometer) were allowed before the test [27].

Testing protocol: The work rate increased by uniform each minute until the patient’s symptoms start. The patient pedaled at a constant rate of 40–60 RPM, unloaded (0 W) for three minutes, and then an increment of 5–10 W per minute for ten minutes was allowed. The examiner was verbally encouraging the patient after the anaerobic threshold (AT) to remain motivated. Cardiopulmonary parameters were measured continuously each minute. Three minutes of unloaded cycling was allowed at the end of the test for cooling down [27]. 

### 2.5. Intervention

#### 2.5.1. Standard Cardiac Rehabilitation Program (Groups A and B)

All participants were offered a full multidisciplinary cardiac rehabilitation program of 12 weeks. The exercise training component was under supervision. All patients participated in educational and self-management sessions and individually tailored therapies, such as nutritional support or psychological support [10].

All the patients attended a 12-week program of aerobic exercise rehabilitation (Intensity: 60–75% MHR, Time: The exercise program consisted of 5 min of warm-up, 30–40 min of moderate aerobic exercises, and 10 min of cool-down, respectively, Frequency: three sessions/week, and Duration: 12 weeks) [28].

#### 2.5.2. Cardiac Rehabilitation Integration with Breathing Training (Group A)

Patients allocated to this form of rehabilitation performed additional breathing training (BE) through two techniques: inspiratory muscle training (IMT) and breathing calisthenics (BC). Their descriptions are as shown in Table 1.

IMT is a device used for the training of the respiratory muscles [7]. Its use has been recommended for many cardiorespiratory conditions including heart failure [17]. It was found that IMT improves respiratory muscle function, aerobic capacity, and quality of life in patients with CHF [23]. However, many studies recommended the use of IMT for CHF patients with prominent respiratory symptoms. It is still not a fundamental component in the cardiac rehabilitation protocols for CHF. 

### 2.6. Outcome Measures

#### 2.6.1. Primary Outcomes 

Respiratory outcomes: [maximal inspiratory pressure (MIP) and rating of perceived exertion (RPE).

#### 2.6.2. Secondary Outcomes 

Cardiovascular [heart rate variability]: high frequency (HF), low frequency (LF), and low frequency to a high-frequency ratio (LF/HF). Resting heart rate (HR), resting systolic blood pressure (SBP), and resting diastolic blood pressure (DBP)].

Cardiopulmonary: maximal oxygen capacity (VO_2_max), anaerobic threshold (AT), maximum heart rate (MHR), and maximum minute ventilation (VE).

### 2.7. Statistical Methods and Sample Size Calculations

Data were expressed as mean ± SD. An unpaired t-test was used to compare between subjects’ characteristics of the two groups. MANOVA was performed to compare within and between the group’s effects for all the measured variables. A statistical package for the social sciences computer program (version 20 for Windows; SPSS Inc., Chicago, IL, USA) was used for data analysis. *p* less than or equal to 0.05 was considered significant.

The sample size was calculated using the G*Power software (version 3.0.10). F-test MANOVA within and between the interaction effects was selected. Considering a power of 0.80, an α level of 0.05 (two-tailed), and an effect size of 0.46; two groups, a generated sample size of at least 20 participants per group was required and a total sample size of 40 subjects [30].

## 3. Results

Data were screened for normality assumption, the homogeneity of variance, and the presence of extreme scores. Shapiro–Wilk and Kolmogorov–Smirnov tests for normality showed that all measured variables are normally distributed. 

No significant difference between the groups was found in age (*p* = 0.77), weight (*p* = 0.09), height (*p* = 0.92), BMI (*p* = 0.75), or EF (*p* = 0.88) (Table 2).

### 3.1. Respiratory Outcomes

There was a significant difference in the mean values of MIP and RPE post-treatment between the two groups (*p* = 0.001 and 0.039), respectively, in favor of group A. There was a significant increase in MIP within group A and a significant decrease in the mean values of RPE within both groups A and B (*p* = 0.001), as shown in Table 3.

### 3.2. Cardiovascular Outcomes 

There was a significant difference between the two groups in the mean values of HF, RHR, RSBP, and RDBP post-treatment (*p* < 0.05) in favor of group A. There was a significant difference between HF, LF/HF, RHR, and RSBP within both groups; HF increased, whereas LF/HF, RHR, and RSBP decreased. Regarding RDBP, there was a significant difference within group A only (*p* = 0.001). No significant changes between groups were noticed in LF post-treatment (*p* = 0.460) and within both groups A and B (*p* = 0.074 and 0.140), respectively as shown in Table 3.

### 3.3. Cardiopulmonary Outcomes

The results of the present study showed a significant increase in the parameters of the functional capacity indices (VO_2_max and AT) and the VE post-treatment in favor of group A (*p* < 0.05). Additionally, there was a significant difference in these variables within group A; VO_2_max, AT, and MHR increased, and VE decreased significantly (*p* < 0.05). There was a significant increase in AT within group B (*p* = 0.007). There was a non-significant difference in MHR between both groups (*p* = 0.730), as shown in Table 4.

## 4. Discussion

The presence of respiratory symptoms in CHF is linked to the mortality risk [31], cardiovascular morbidities, and exercise capacity of the patients [32]. 

CR is a globally accepted and standard treatment for CHF. However, the research has limited information about the effect of addressing respiratory symptoms through standard CR. 

Respiratory muscle abnormalities are highly prevalent in CHF and are associated with an increase in the ergoreflex. This results in increased work of ventilation and contributes to the increased sympathetic activation of the heart failure syndrome, causing the sensation of fatigue and breathlessness [6,15].

The present study showed that MIP and RPE, HF, RHR, RSBP and RDBP, VO_2_max, AT, and VE were improved significantly post-treatment in the group that performed BE in addition to CR better than in the CR group. There were no differences in the HR max, LF, and LF/HF post-treatment between both groups. 

### 4.1. Respiratory Outcomes

The results in the current study showed a post-treatment increase in the parameters of inspiratory muscle strength (MIP) by 20.6% and 3%, and dyspnea scores (RPE) by 20.9% and 13.6% in groups A and B, respectively. 

The results of the current study coincided with the recommendations of a recent systematic review by Sadek et al. [29], which described the recommended IMT protocol for CHF patients. The recommended program was to train patients at 60% of the MIP, six sessions/week for 12 weeks. They also concluded that this would result in a marked improvement in inspiratory muscle strength, walking distance, and dyspnea. The IMT protocol used in the current study is similar to the dose recommended in this systematic review. Additionally, the outcomes reported in this study were flowing well with the review’s conclusions. 

However, the same authors mentioned above (Sadek et al. [30]) performed an RCT after 2 years which still align with the concept in our study that IMT may be a complementary therapy in patients with CHF; the percentages of improvement were quite far away from our reported results. In their RCT, the authors examined the influence of 12 weeks of IMT together with aerobic training in CHF patients. The experimental group increased inspiratory muscle strength by 62% (versus 20.6% only in our study). The exercise frequency utilized in their study was three sessions/wk (versus six sessions/wk in our study). 

After careful investigation, it was found that the main difference between this RCT and the current one is the exercise frequency per week. In the current study, we followed the exercise recommendations mentioned in the systematic review mentioned previously [29]. Consequently, we performed six sessions of IMT per week, and that might be the reason that we have poorer results in comparison with Sadek et al.’s RCT. That might suggest that the increased frequency of IMT use per week in CHF patients might lead to fatigue and/or exhaustion, decreasing the amount of benefit. Such findings have to be investigated in future studies to construct clear recommendations about the exercise prescription of IMT that best suits CHF patients.

### 4.2. Cardiovascular Outcomes

There were increases in the post-treatment parameters of HRV (HF by 40, 27.7%, LF by 2, 1.4%, LF/HF ratio by 29.4, 23%), RHR by 9.6, 5.1%, RSBP by 7.6, 4.2%, and RDBP by 4.5, 1% in groups A and B, respectively.

That came in agreement with a study carried out by Catella et al. [33] who examined the effect of slow breathing on the autonomic system through the assessment of baroreflex sensitivity. The authors concluded that baroreflex sensitivity can be enhanced significantly by slow breathing, both in healthy conditions and in the presence of CHF, resulting in a reduction in heart rate and arterial blood pressure. 

However, the results of the present study showed a non-significant reduction in the low-frequency component of HRV (LF) in group A. Previous investigators reported that the interpretation of the LF component is more controversial, as it represents a mixture of the sympathetic and parasympathetic modulation of the heart rate, yet the exact contribution of each component to LF is unknown [34].

### 4.3. Cardiopulmonary Outcomes

The results of the current study showed a post-treatment increase in the parameters of VO_2_max by 10.5, 3%, AT by 18.3, 6.6%, VE by 12.9, 0.4%, and MHR by 4, 2% in groups A and B, respectively.

This came in agreement with Lopes et al. [35] who evaluated the effects of breathing techniques on functional capacity, ventilatory responses to exercise, recovery oxygen uptake kinetics, and quality of life in 33 patients with CHF and inspiratory muscle weakness (maximal inspiratory pressure [MIP] <70% of predicted). The IMT resulted in an increase in the peak oxygen uptake and the 6 min walk distance. Likewise, circulatory power (calculated as the product of Peak VO2 and Peak systolic pressure) increased, and ventilatory oscillations were reduced. 

On the other hand, the percentage of change in the MHR is less than that reported with the VO_2_max among group A. This shows that the effect of inspiratory muscle training in patients with CHF is largely attributed to central [as the VO_2_ max is limited to cardiac output (VO_2_ max = arterio-venous O_2_ deference multiplied by cardiac output)] rather than peripheral (the ability of a muscle to move against a larger workload). This finding may not be surprising as it could be related to the absence of the effect of whole-body training and the conditioning effect of aerobic exercise [36].

### 4.4. Mechanisms

One of the possible explanations for the above-mentioned results is that the IMT can delay the development of diaphragmatic fatigue in patients with CHF [2,3], leading to a reduction in the recruitment of accessory respiratory muscles [15]. This reduces the blood flow required by the respiratory muscles during exercise [4].

In addition, in CHF, there is enhanced ventilatory sensitivity to both central and peripheral chemoreceptor stimulation, which can reflexively induce sympathetic vasoconstriction and reduce blood flow to skeletal muscles during exercise [14]. 

Therefore, diaphragm fatigue may elicit sympathetically mediated vasoconstriction in limb muscles, a reflex that might be attenuated by inspiratory muscle training which delayed the development of fatigue of the diaphragm [6]. This consequently reduces sympathetic activation, improves the perfusion of the peripheral muscles [15], and affects cardiovascular, cardiopulmonary, and respiratory outcomes. 

Furthermore, the significant improvements reported in anaerobic threshold during cardiopulmonary exercise may be attributed to the significant reduction in ventilation (VE) in group A. This came in parallel with the study of Audrey et al. [37] in which reducing the work of breathing using a proportional assist ventilator during cycling exercise increased the blood flow to the legs. An average 50% reduction in the work of breathing results in a 5–7% increase in leg blood flow. 

It is believed that the significant reduction in VE seen in the present study is partially attributed to the shifting of the respiratory muscle metabolism from a catabolic to an anabolic state secondary to training. This may result in the attenuation of ergoreceptors originating from this muscle, explaining the hyperventilation response in CHF. 

Therefore, improvement in ventilatory muscle strength and endurance may also attenuate the exaggerated ergoreceptors. The techniques improve not only ventilatory manifestations but also cardiovascular and cardiopulmonary outcomes that may be affected by the ergoreflex activation and sympathetic overactivity usually experienced in CHF [6]. 

Limitations: This study has mainly a limitation about the accessibility to the full medical history of the participants (etiology of CHF, family and smoking history, other co-morbidities) due to the nature of the clinical practice in Egypt. Further studies with the consideration of this confounder will be conducted in the future in a better-structured RCT.

## 5. Conclusions

Breathing training using IMT and BC for 12 weeks in combination with CR may have a positive impact on the respiratory, cardiovascular, and cardiopulmonary outcomes in patients with CHF more than standard CR.

CR programs have to be individually tailored as mentioned in the European and American guidelines [38,39]. As the respiratory symptoms are patient-centered concern for CHF, it is highly recommended to design CR targeting these symptoms and considering them among the program outcomes. 

## Figures and Tables

**Figure 1 ijerph-19-14694-f001:**
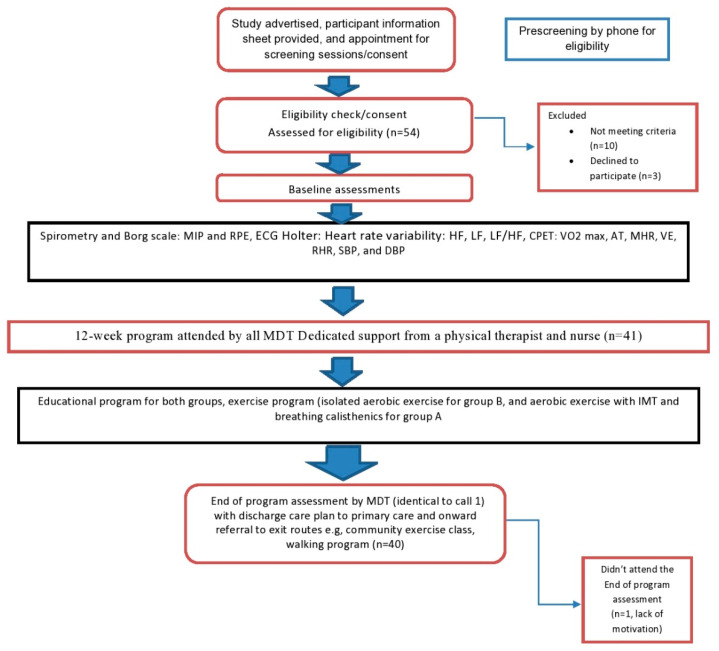
Design of the study.

**Table 1 ijerph-19-14694-t001:** IMT and breathing calisthenics description.

Points	Inspiratory Muscle Training (IMT)	Breathing Calisthenics
Definition	IMT is a device used for respiratory muscle training to improve the function of the respiratory muscles through specific exercises. The IMT device (Respironics, Chichester, UK) is very easy to use and practical and does not require large amounts of money.	A series of breathing exercises performed as eight repetitions of each exercise were completed. These exercises strengthened the abdominal muscles
Method of use	The patient who utilized this IMT was asked to inspire deeply through the mouthpiece of it against the selected load with all prescribed medications.	In the supine position, the legs were flexed to the abdomen as the patient exhaled (abdominal knee bend exercise). In a separate exercise, the head and shoulders were raised from the mat as the patient exhaled (abdominal sit-up exercise). In a third exercise, the patient was seated. He was asked to clench both hands behind the head and to move them back and forth during inspiration and expiration, respectively pulling his belly inward with each inhalation.
Intensity	The initial workload is measured as 30% and increased gradually to 60% of MIP. Subject trains in the initial workload for 2 weeks, then the target workload was increased by 5% every 2 weeks	The training intensity was adjusted over the weeks by using the sensation of dyspnea as a parameter, keeping it between 4 and 6 on the Borg CR 10 scale
Duration	Sessions were divided into six sets (12 weeks) five minutes in duration, and separated by 5 min of rest.	Sessions were divided into three sets, eight repetitions of each exercise, separated by 5 min of rest in between sets.
Frequency	six times/week [26]	six times/week [29]

**Table 2 ijerph-19-14694-t002:** Mean ± SD of initial demographic and clinical characteristics of the studied groups.

Variable	IMT Group	Control Group	MD	t-Value	*p*-Value
	X ± SD	X ± SD			
Age (year)	55.55 ± 8.36	55.90 ± 6.54	0.40	1.49	0.883
Height (cm)	174 ± 12.49	173 ± 10.49	1.1	0.102	0.919
Weight (kg)	85.650 ± 17.623	79.45 ± 20.43	6.2	1.764	0.094
EF (%)	33.10 ± 4.52	33.30 ± 4.54	0.20	0.135	0.880
NYHA	2.65 ± 0.48	2.50 ± 0.51	0.15	1.00	0.330

**Table 3 ijerph-19-14694-t003:** Comparison between and within groups pre- and post-study for mean values of respiratory and cardiovascular outcomes.

Respiratory and Cardiovascular Outcomes	Pre-Study Mean ± SD	Post-Study Mean ± SD	% of Change	*p*-Value
MIP				
Group A	35.9 ± 2.7	43.3 ± 3.6	20.6%	0.001 *
Group B	35.6 ± 2.9	36.7 ± 3.3	3%	0.277
*p*-value	0.727	0.001 *		
RPE				
Group A	4.3 ± 0.5	3.4 ± 0.6	20.9%	0.001 *
Group B	4.4 ± 0.6	3.8 ± 0.6	13.6%	0.001 *
*p*-value	0.584	0.039 *		
LF				
Group A	66.3 ± 2.5	65.1 ± 2	2%	0.074
Group B	66.5 ± 1.9	65.6 ± 1.6	1.4%	0.140
*p*-value	0.674	0.460		
HF				
Group A	6.5 ± 0.6	9.1 ± 1.4	40%	0.001 *
Group B	6.5 ± 0.8	8.3 ± 1	27.7%	0.001 *
*p*-value	0.999	0.013 *		
LF/HF				
Group A	10.2 ± 1.1	7.2 ± 1.1	29.4%	0.001 *
Group B	10.4 ± 1.4	8 ± 1.1	23%	0.001 *
*p*-value	0.766	0.056		
RHR				
Group A	86.1 ± 4	77.8 ± 4.7	9.6%	0.001 *
Group B	85.8 ± 4.9	81.4 ± 5.3	5.1%	0.004 *
*p*-value	0.868	0.019 *		
RSBP				
Group A	136.8 ± 4.3	126.4 ± 5.1	7.6%	0.001 *
Group B	136 ± 4	130.3 ± 5	4.2%	0.001 *
*p*-value	0.564	0.009 *		
RDBP				
Group A	81.8 ± 2.9	78.1 ± 2.4	4.5%	0.001 *
Group B	82 ± 2	81.1 ± 2	1%	0.250
*p*-value	0.827	0.001 *		

SD: standard deviation *p*-value: probability value, MIP: maximum inspiratory pressure, RPE: rating of perceived exertion, LF: low frequency, HF: high frequency, RHR: resting heart rate, RSBP: resting systolic blood pressure, RDBP: resting diastolic blood pressure, *: significant.

**Table 4 ijerph-19-14694-t004:** Comparison between and within groups pre- and post-study for mean values of cardiopulmonary outcomes.

Cardiovascular Outcomes	Pre-Study Mean ± SD	Post-Study Mean ± SD	% of Change	*p*-Value
VO_2_ max				
Group A	14.3 ± 1.4	15.8 ± 1.3	10.5%	0.001 *
Group B	14.5 ± 1.2	14.9 ± 1.2	3%	0.307
*p*-value	0.735	0.045 *		
AT				
Group A	51.3 ± 1.8	60.7 ± 6.5	18.3%	0.001 *
Group B	51.1 ± 1.9	54.5 ± 3.2	6.6%	0.007 *
*p*-value	0.936	0.001 *		
MHR (beat/min)				
Group A	118.3 ± 6.6	123.1 ± 5.1	4%	0.013 *
Group B	120 ± 6	122.4 ± 5.8	2%	0.205
*p*-value	0.354	0.730		
VE				
Group A	46.4 ± 1.2	40.4 ± 1.4	12.9%	0.001 *
Group B	46.2 ± 1.3	46 ± 1.4	0.4%	0.657
*p*-value	0.729	0.001 *		

SD: standard deviation, *p*-value: probability value, VO_2_ max: maximum oxygen capacity, AT: anaerobic threshold, MHR: maximum heart rate, VE: Maximum minute ventilation, *: significant.

## Data Availability

Not applicable.

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
