# Peer review of "The Need for Breathing Training Techniques: The Elephant in the Heart Failure Cardiac Rehabilitation Room: A Randomized Controlled Trial"

_ijerph, 2022, doi:10.3390/ijerph192214694_

Round 1
Reviewer 1 Report
Overview: A very interesting study and well designed. The only problem I had with the manuscript was the usage of too many one sentence paragraphs. However, I only noted a few instances because I suspect that it’s how the journal works in setting up papers.
Specific Recommendations
Page 1, Line 46 – Should be “CHF” not “chronic heart failure”.
Page 2
Line 51 – Change to “Typically, standard CR programs do not include strategies to control respiratory symptoms [7], [8]”
Line 53 – Change to “The most successful state-of-the-art CR programs globally like the EuroAction[9], Our hearts Our minds[10], [11], Coroprevention[12], and the Million hearts [13] do include CHF among their targeting population.
Line 55 – “However, none of them consider……”
Lines 62 to 64 – Eliminate the “are” after (BE) and combine this one sentence paragraph to the pone above it.: “Breathing Training (BE) techniques that aim to improve the function of the respiratory 6muscles through specific exercises incorporating different methods like inspiratory Muscle Training (IMT) and breathing calisthenics (BC).[16].
Lines 62-72 – Combine the last three paragraphs into a single paragraph.
Line 71 – Delete this sentence. It is not needed. “This is in a way to deliver- and then recommend the design of- an individually-tailored CR programs to CHF patients.”
Page 4 line 114 – Combine this sentence to the paragraph above it.
“MIP was performed to measure inspiratory strength before and after the study for both groups 114 and to determine the starting load for the inspiratory muscle training group.”
Page 5
Line 116 – Combine these 2 sentences to the paragraph below them.
“Before the measurement, all the procedures were explained in detail to the patient. Therefore, the participants……..”
Line 127 – Combine all of these sentences into a single paragraph.
“2.4.3. Hours Ambulatory Electrocardiography (ECG) recording (Holter)”
Page 9 line 242 – What does “stilalignsgn” represent?
“However, the same authors mentioned above (Sadek et al.,[31] ) did an RCT after 2 years which stilalignsgn with the concept. “
You used this phrase too many times. “Results in the current study”
Author Response
We found the comments very useful and agreed to implement all changes regarding editing and merging paragraphs. All changes can be tracked with “track changes feature.
Reviewer 2 Report
I am interested in the study entitled “Physical exercise interventions in heart failure rehabilitation programs have to be patient-centered: Future perspective towards a personalized treatment.” By Abeer Farghaly et al.
Authors showed the beneficial effect of breathing exercise to the CHF patients, I am interested the data. I raise several points outlined below.
I think the title is not suitable for the conclusion. I think authors were analyzed the effect of the breathing exercise, did not mentioned as to the physical exercise and personalized treatment.
In the abstract, authors wrote as only “IMT and BC”, I think it is difficult for readers to understand, and the abbreviation may be better to be explained in this part.
Figure 1 is difficult to understand, especially bottm half of the figure.
In the method of “Cardiac rehabilitation integration with breathing training (2.5.2), I think authors should show the device in detail, and related literatures.
In the method of Primary outcome (2.6.1), I think authors should show the measuring method in detail and should show the related literatures.
In the method of Primary outcome (2.6.2), I think authors should show the measuring method in detail and should show the related literatures. Especially, what software used , the definicion of LF and HF as to frequency, what part of the ECG data were used, all day or only part of resting time, and how to dealed arrhythmias and motion noise. I think authors should describe the method in detail.
In the Table 2, if possible, I want to know the more data as to the patients background, for example, etiology of CHF, hypertension, diabetes mellites, dyslipidemia, obesity and smoking, BNP or NT-proBNP, medication etc.
In the table 3 and 4, I think the abbreviation may be better to be explained in the table legends. And I think authors should explain the “RHR, “RSBP” and “RDBP”, I am sorry, I could not find the explanation of these abbreviation.
In the results, I think authors should show the detailed data as to the breathing exercise, what percentage of patients could underwent BE program, and authors should show the any adverse event with BE program.
In the results of cardiovascular outcomes, if possible, I want to know the change of cardiac function, for example LVEF, BNP etc.
Author Response
Thank you for your tremendous efforts. We really appreciate the time and effort that you and the other reviewers have dedicated to review our manuscript. The comments were encouraging and you provided very useful suggestions and recommendations which helped us to improve the quality of the manuscript. Recommendations were considered and applied to the final version.
Comment 1: I think the title is not suitable for the conclusion. I think authors were analyzed the effect of the breathing exercise, did not mentioned as to the physical exercise and personalized treatment.
Response: Thank you for this comment. We actually added the breathing exercise to the new suggested title.
Comment 2 Write the full words of IMT and BC in abstract
Response: the full words Inspiratory Muscle training was added to the abstract
Comments 3 Figure 1 is difficult to understand, especially bottom half of the figure.
Response: We have edited the figure and simplified it as much as we could.
Comments 4: In the method of “Cardiac rehabilitation integration with breathing training (2.5.2), I think authors should show the device in detail, and related literatures.
Response : the type of IMT device was added and related literature was added
Comments 5: In the method of Primary outcome (2.6.1), I think authors should show the measuring method in detail and should show the related literatures.
In the method of Primary outcome (2.6.2), I think authors should show the measuring method in detail and should show the related literatures. Especially, what software used , the definition of LF and HF as to frequency, what part of the ECG data were used, all day or only part of resting time, and how to dialed arrhythmias and motion noise. I think authors should describe the method in detail.
Response : Details of primary outcomes 2.6.1 in lines 119-128 and 2.6.2 , were added in 2.4.1. and 2.4.3 respectively.
Comments 6: In the Table 2, the more data as to the patients' background, for example, etiology of CHF, hypertension, diabetes mellitus is not available.
Response: Unfortunately, this data is not available. That will be added to the limitations section.
Comments 7: In the table 3 and 4, I think the abbreviation may be better to be explained in the table legends. And I think authors should explain the “RHR, “RSBP” and “RDBP”, I am sorry, I could not find the explanation of these abbreviation.
Response: legends with all abbreviation were added under table 3 and 4, and these abbreviations were added to the abbreviation list.
Comments 8 : In the results, I think authors should show the detailed data as to the breathing exercise, what percentage of patients could underwent BE program, and authors should show the any adverse event with BE program.
Response : 50% of the patients underwent BE (Group B), and no adverse effect were reported regarding BE techniques
Comments 9: In the results of cardiovascular outcomes, if possible, I want to know the change of cardiac function, for example LVEF, BNP etc
Response: In the results of cardiovascular outcomes: LVEF is among the echocardiographic changes that might enrich the data for sure, however, that was not among our scope in this study. Therefore, it was not measured. We think it can be measured in a further study.
Please note that all revisions made to the manuscript were marked up using the
“Track Changes”
We look forward to hearing from you regarding our submission and to responding to any further questions and comments you may have.
Best regards
Hady Atef
The Corresponding author